# Maternal obesity and prenatal alcohol exposure are associated with child development: Results from the Safe Passage Study

Ayesha Sania[1]*, Shreya Rao[1], Nicolò Pini[1], Mandy Potter[2], Yael Rayport[1], Liana Eisler[1], Lucy Brink[2], Jyoti Angal[3,4], Michael M. Myers[1], Hein Odendaal[2], Amy J. Elliott[3,4], William P. Fifer[1,5,6], Lauren C. Shuffrey[7]

1 Department of Psychiatry, Columbia University Irving Medical Center, New York, United States of America, 2 Department of Obstetrics and Gynaecology, Faculty of Medicine and Health Science, Stellenbosch University, Cape Town, Western Cape, South Africa, 3 Avera Research Institute, Sioux Falls, South Dakota, United States of America, 4 Department of Pediatrics, University of South Dakota Sanford School of Medicine, Sioux Falls, South Dakota, United States of America, 5 Division of Developmental Neuroscience, New York State Psychiatric Institute, New York, United States of America, 6 Department of Pediatrics, Columbia University Irving Medical Center, New York, United States of America, 7 Department of Child and Adolescent Psychiatry, NYU Grossman School of Medicine, New York, United States of America

☯ These authors contributed equally to this work.
* as4823@cumc.columbia.edu

## Abstract

A large body of evidence supports the role of the prenatal environment in shaping childhood development. The relative contributions of prenatal alcohol use (PAE), maternal socioeconomic, and nutritional status on child development vary in high-versus low-income settings. We analyzed data from a prospective cohort study among mother-infant dyads from Cape Town (CT), South Africa and the Northern Plains (NP), USA. The Mullen Scales of Early Learning were administered by trained assessors to evaluate cognitive, motor, and language development of 1-year old children. We used multiple linear regression models to assess standardized mean differences in development scores by (1) maternal prenatal factors, (2) delivery factors and (3) child factors within each study site. 1,728 infants from CT and 1,140 infants from the NP were included in the analyses. In CT, infants with moderate-to-high PAE had 0.17 SD (95% CI −0.30, −0.04) lower cognitive and 0.15 SD (−0.29, −0.2) lower expressive language scores compared to infants without PAE. In the NP, maternal obesity (BMI > 30 kg/m$^2$) was significantly associated with −0.21 SD (−0.36, −0.06), and −0.13 SD (−0.27, −0.02) reductions in cognitive, and expressive language scores, respectively. Household crowding, lower levels of maternal educational attainment, prenatal maternal depression, low birthweight, admission to neonatal intensive care unit, and male sex had significant negative associations with cognitive and language development in both sites with effects ranging from −0.32 to −0.11 SDs. These results highlight the importance of assessing risk factors by populations across

**Data availability statement:** De-identified data from the Safe Passage Study is available through NICHD's Data and Specimen Hub (DASH). All cases of demographic and exposure data are available on DASH. Elliott, Amy (2025). A Prospective Study on the Role of Prenatal Alcohol Exposure in SIDS and Stillbirth (Version 1). NICHD Data and Specimen Hub. https://doi.org/10.57982/sv8c-4y07. The tribal data used in this study is restricted access per the requirements of participating tribal nations and the Indian Health Service IRB. Avera Health maintains the data on a secure server, and people can contact Dr. Christine Hockett (Christine.hockett@avera.org) to learn the process for gaining tribal approval and the necessary regulatory approvals to gain access.

**Funding:** This research was supported by grants UH3OD023279, U01HD055154, U01HD045935, U01HD055155, and U01AA016501, issued by the Office of the Director of the National Institutes of Health, National Institute on Alcohol Abuse and Alcoholism, Eunice Kennedy Shriver National Institute of Child Health and Human Development, and the National Institute on Deafness and Other Communication Disorders. Ayesha Sania was supported by a career development award from the Fogarty International Center at the NIH (1K01TW012425-01A1). The opinions expressed in this paper are those of the authors and do not necessarily represent the official views of the National Institutes of Health, the Eunice Kennedy Shriver National Institute of Child Health and Development, or Fogarty International Center, the National Institute on Alcohol Abuse and Alcoholism, or the National Institute on Deafness and Other Communication Disorders.

**Competing interests:** The authors have declared that no competing interests exist.

diverse social and cultural environments and emphasize the imperative to formulate intervention packages tailored to the local context.

## Introduction

Understanding the predictors of early childhood development to support policy and intervention programs is a key priority worldwide, particularly in low-to-middle income countries (LMICs) where the majority of the world's children reside. According to recent estimates, a third (80 million) of the 3 and 4-year-old children in LMICs do not attain their full developmental potential [1]. Early cognitive, language, and socioemotional development are key determinants of future educational and income achievement in adult life [2]. Suboptimal development in the early childhood period can have a lasting impact on the life course of individuals as well as the larger society. Therefore, understanding the independent contributions prenatal exposures on child developmental outcomes is a significant global public health priority.

The prenatal period is a critical developmental window that has the potential to shape long-term health outcomes for offspring through fetal programming mechanisms. A growing body of research within the developmental origins of health and disease (DOHaD) framework highlights how adverse in-utero environmental factors including poor maternal nutritional status, exposure to harmful chemicals, crowded living conditions, and socioeconomic challenges impact children's cognitive, motor, and language development [3–6]. The impact of risk factors on child development varies between high- and low-income settings, reflecting important contextual differences in exposure patterns, socioeconomic status, and postnatal environments. For example, research from low-income settings consistently reports harmful effects of prenatal alcohol exposure (PAE) on both cognitive and motor skills [7–10], whereas the same patterns are not observed in high-income contexts [11–13], which suggests positive environmental factors in high-income settings might buffer the impact of PAE. Albeit, in high income settings, epidemiologic evidence demonstrates clear neurodevelopmental harm associated with prenatal alcohol exposure at moderate to higher levels [14,15]. Apparent protective effects of mild drinking are potentially explained by the confounding effects of maternal education and favorable socioeconomic conditions [16–18]. In addition, the null effects of PAE on child development in high income settings could be due to an enriched postnatal environment which acts as a buffer to the harmful effects of PAE on child development.

The role of prenatal maternal nutritional status (maternal anemia, short stature, and low body mass index [BMI] during pregnancy) [19,20] on child development is well established. While studies from LMICs report association of low BMI during pregnancy with poor child development, recent studies report maternal obesity (BMI > 30 kg/m2) as an emerging risk factor for poor child health and development primarily in high-income settings [21–24]. Recent systematic reviews of studies from high income countries report association of maternal obesity with lower cognitive

development scores, developmental delay, attention deficit hyperactive disorder (ADHD, and autism spectrum disorder (ASD) [25,26]. Given the increasing rates of obesity worldwide, further studies are needed to examine this relationship in both high- and low-income settings.

In the present study, we estimated the associations of several prenatal exposures including PAE, maternal obesity, delivery and birth characteristics, and socioeconomic characteristics, with children's motor, cognitive, and language development in two large socioeconomically, ethnically, and culturally diverse cohorts of mother-infant dyads from the USA and South Africa using data from the NIH Safe Passage Study [27]. Estimating the relative contributions of known and emerging risk factors, while considering contextual factors in both high-income and LMICs, can help inform programs and policies aimed at targeting interventions for at-risk infants.

## Methods

### Study population

Participants were enrolled in the Safe Passage Study by the Prenatal Alcohol and SIDS and Stillbirth (PASS) network which was designed to examine the effects of prenatal drinking and smoking on adverse birth outcomes [27]. This multi-site study included data from Western Cape Province, South Africa and five sites from the Northern Plains, United States, three of which were on American Indian Reservations. In South Africa, participants were recruited from the Bishop Lavis and Belhar residential areas in and around Cape Town, serving predominantly individuals of mixed ancestry. In the United States, sites in North Dakota and South Dakota represented rural and semi-urban communities, including tribal communities. Women were eligible to participate in the study if they were 16 years or older, pregnant with one or two fetuses with a gestational age of 6 weeks or more, spoke English or Afrikaans, and were not planning to relocate from the catchment area during the study period. The study design has previously been described in detail by Dukes et al [27]. Between August 2007 and January 2015, approximately 12,000 women enrolled in the study, of them a subset of 3750 women were selected for additional study assessments during postnatal period. At the time of the recruitment interview, approximately 1 in 3 eligible and consenting women who are less than 24 weeks gestation were selected at random to participate in the embedded study in which child developmental assessments were administered. The analysis presented in this manuscript includes all singleton children with early childhood development assessment data from the Northern Plains (NP), USA and Cape Town (CT), South Africa. The study flow chart details the selection of the final analytic sample (S1 Fig).

### Developmental assessment

Child development was assessed with the Mullen Scales of Early Learning (MSEL) when the children were 1 year old. Infants born prematurely (<37 weeks gestation) were assessed at their adjusted postnatal age of 1-year. The Mullen is standardized play-based developmental assessment that assesses fine motor, gross motor, visual reception, expressive language, and receptive language function standardized to a T distribution. The cognitive T score in the MSEL is derived from the sum of visual reception, fine motor, receptive language, and expressive language subscales. The Mullen has been adapted to and validated in Afrikaans to South African context [28]. The Mullen has also been used to assess the developmental trajectories of Native American children in the Northern Plains [29]. In both locations, trained research staff administered the Mullen assessments. Assessors administering the Mullen evaluations were not involved in exposure assessment and were unaware of pregnancy and delivery conditions.

### Assessment of risk factors

**Socioeconomic data.** Information on marital/partner status was collected at the time of enrollment, and household crowding (ratio of number of people per room in a household) was collected at the first prenatal visit.

**Maternal-Infant chart abstraction.** Data on maternal medical diagnoses, birth and delivery characteristics including gestational diabetes mellitus, hypertension, preeclampsia, induced or augmented labor, cesarean section, infant resuscitation, NICU admission, infant sex, and birthweight were collected from chart abstraction by research midwives.

**Maternal anthropometrics.** Maternal height and weight were collected at the enrollment visit. The median gestational age at enrollment was 20 weeks (inter quartile range, IQR: 14, 25) in Cape Town and 16 weeks (IQR: 13, 18) in Northern Plains.

**Alcohol and tobacco use during pregnancy.** Prenatal drinking and smoking information was collected longitudinally during up to four prenatal visits. A 30-day timeline follow-back interview was used to assess prenatal alcohol and tobacco exposure [30]. During up to four prenatal visits, participants reported the day they most recently drank or smoked, and their substance use for the 30-days prior to it. They also described the quantity, sharing, and frequency of alcohol and cigarette consumption. The information was used to estimate the standard number of drinks of alcohol consumed per day of pregnancy and the average number of cigarettes smoked per week of pregnancy. Missing drinking and smoking data were imputed using a k-nearest neighbor (kNN) algorithm as described previously [31]. Daily alcohol and weekly smoking estimates were then aggregated within trimester to derive trimester-level averages used for clustering analysis. Using a finite mixture model–based clustering approach previously described [32], participants were classified into four smoking groups and ten drinking groups based on the amount and timing of exposure during pregnancy. Because several of the drinking groups had a small number of subjects, they collapsed into 4 (non-smoker/drinker, quit early, low continuous, and moderate to high continuous) smoking and drinking groups for prior analyses [33,34] and the current analyses.

**Prenatal maternal depression assessment.** Maternal depression was assessed using the Edinburgh Postnatal Depression Scale (EPDS), which has been validated as a screening instrument in South Africa [35]. An EPDS cutoff was used to categorize women into no depression (EPDS<13), probable depression (EPDS 13–15), and major depressive disorder (EPDS >15) groups. Although EPDS thresholds of 10 or 13 are commonly used in U.S. and other high-income settings for perinatal depression screening, we applied this categorization to maintain comparability across study sites [36].

## Statistical analysis

To ensure comparability between the measurement in the two study sites, all development outcome scores were standardized (z-scored). Within each study site, linear regression models were used to assess standardized mean differences (SMDs) in cognitive, fine motor, gross motor, expressive language, and receptive language scores for the selected risk factors. For each outcome and site, we first fit minimally adjusted models including each risk factor separately, adjusting for child age and sex. Risk factors meeting a threshold of $p < 0.20$ in these models were subsequently included in the multivariable model for that outcome. Because variable selection was conducted separately for each outcome and site, the final multivariable models included slightly different combinations of risk factors.

We used separate models for three sets of risk factors based on their timing of measurement during the study. This allows us to avoid introduction of bias in our analyses by adjusting for variables on the hypothesized causal pathway between a risk factor and child development. The three models with different sets of predictors are: (1) model 1 included maternal socio-economic, demographic, drinking, smoking, and nutritional characteristics; (2) model 2 included pregnancy complications such as hypertensive disorders of pregnancy, gestational diabetes, and maternal socio-economic indicators; (3) model 3 included delivery characteristics including location of delivery, birthweight, need for resuscitation and NICU admission and maternal socio-economic indicators. All multivariable models included child's biological assigned sex at birth and age at assessment. Missing values for covariates were replaced with a "missing" indicator and included in the multivariable models. Because we have related outcomes and covariates,

adjustment for multiple comparisons were not appropriate [37–39]. All analyses were performed in SAS software version 9.4 (SAS Institute, Cary NC).

### Ethical approval

Written informed consent for the Safe Passage Study and a separate informed consent for the follow-up assessments were obtained from all participating mothers. Ethical approval for the Safe Passage study was acquired from the Health Research Ethics Committee of Stellenbosch University in South Africa, and Sanford Health Institutional Review Board, the Indian Health Service, participating Tribal Nations, and the Institutional Review Board of the New York State Psychiatric Institute in the United States.

## Results

The final analyses included 2868 infants, of them 1728 (60%) were from Cape Town (CT) and 1140 (40%) were from the Northern Plains (NP). Table 1 shows the characteristics of the study participants by site. Overall, participants from CT had poorer socioeconomic conditions, as reflected in their lower educational attainment, higher crowing index, less frequent access to antenatal care, and higher unemployment rates. Women in the NP were taller, and a greater proportion were overweight and obese. In CT, 23% women continued smoking in moderate to high amount throughout the pregnancy compared to only 4% woman in the NP. Women in the moderate-to-high continuous group smoked 48.31 (SD 21.71) cigarettes/week, on average in first trimester. The majority (43%) of NP participants quit drinking in the first trimester, only 6% drank moderate-to-high amount, while 19% participants from CT continued to drink in moderate to high amount throughout pregnancy. Women in the high moderate-to-continuous group drank an average total drink of 40. 9 (SD 60.1) drinks in first trimester (S1 Table).

### Predictors of child development in Cape Town

The estimates of the association of predictors with infant development scores in Cape Town are presented in Figs 1–5. PAE was associated with reduced cognitive and expressive language scores. Infants with moderate to high levels of PAE had 0.17 SD (95% Confidence Interval, CI: −0.30, −0.04) lower cognitive scores and 0.15 SD (CI −0.29, −0.02) lower expressive language scores, as compared with infants with no PAE. Indicators of SES such as maternal education and crowding index, markers of socioeconomic conditions, were associated with expressive, language and gross motor scores. In comparison to infants of mothers with more than a high school education, infants of mothers with primary school education had 0.35 SD (CI: −0.64, −0.06) lower expressive language scores and 0.33 SD (CI: −0.2, −0.03) gross motor scores. Infants of mothers with probable depression (EPDS 13−15) had lower gross motor scores (−0.17 SD, CI: −0.33, −0.02) compared to mothers without depression. However, infants of mothers with major depression (EPDS >15) did not differ significantly in gross motor scores compared with the reference group. With each unit increase in the crowding index, there was 0.07 SD and 0.06 SD reduction in cognitive and expressive language scores, respectively. We found no significant association of parity, number of antenatal care visits, maternal BMI during pregnancy, or marital/partner status with infant developmental outcomes on the MSEL.

Among the child and delivery factors, child sex, low birthweight, and admission to NICU were each independently associated with infant development on the MSEL. Compared to normal birthweight infants, infants born with birthweight below <2000 g had on average 0.46 SD (CI: −0.73, −0.020) lower cognitive and 0.55 SD (CI: −0.85, −0.25) lower fine motor scores. Infants who were admitted to the NICU after birth had 0.26 SD (CI: −0.45, −0.06), 0.28 SD (CI: −0.49, −0.07), and 0.31 SD (CI: −0.49, −0.12) lower cognitive expressive and receptive language scores, respectively. Female infants had on average higher cognitive, expressive, and receptive language scores compared to male infants. We did not find any significant associations between resuscitation at birth with any development domains on the MSEL.

**Table 1. Characteristics of the study population.**

| Characteristics | Cape Town, South Africa (N = 1728) | Northern Plains, USA (N = 1140) | Overall (N = 2868) |
|---|---|---|---|
| **Maternal factors** | | | |
| **Maternal age (years)** | | | |
| Mean (SD) | 25 (± 5.8) | 27 (± 5.1) | 26 (± 5.6) |
| **Height (cm)** | | | |
| <145 | 14 (1%) | 20 (2%) | 53 (2%) |
| 145-150 | 119 (7%) | 7 (1%) | 126 (4%) |
| 150-155 | 348 (20%) | 35 (3%) | 383 (13%) |
| >155 | 1228 (71%) | 1078 (95%) | 2306 (80%) |
| Missing | 19 (1.1%) | | 39 (1.4%) |
| **BMI (kg/m^2)** | | | |
| <18.5 underweight | 133 (8%) | 8 (1%) | 141 (5%) |
| 18.5-25 normal | 856 (50%) | 389 (34%) | 1245 (43%) |
| 25-30 overweight | 361 (21%) | 334 (29%) | 695 (24%) |
| >30 obese | 355 (21%) | 386 (34%) | 741 (26%) |
| Missing | 23 (1%) | 23 (2%) | 46 (2%) |
| **Parity** | | | |
| 0 | 695 (40%) | 411 (36%) | 1106 (39%) |
| 1 | 532 (31%) | 344 (30%) | 876 (31%) |
| 2 | 323 (19%) | 217 (19%) | 540 (19%) |
| 3 or more | 177 (10%) | 168 (15%) | 345 (12%) |
| Missing | 1 (0%) | 0 (0%) | 1 (0%) |
| **Antenatal care visits** | | | |
| <3 | 122 (7%) | 7 (1%) | 129 (4%) |
| 3-6 | 966 (56%) | 53 (5%) | 1019 (36%) |
| >6 | 592 (34%) | 1072 (94%) | 1664 (58%) |
| Missing | 48 (3%) | 8 (1%) | 56 (2%) |
| **Education** | | | |
| Primary school education | 122 (7%) | 12 (1%) | 134 (5%) |
| Some high school education | 1136 (66%) | 165 (14%) | 1301 (45%) |
| High school completed | 382 (22%) | 176 (15%) | 558 (19%) |
| Higher than high school | 85 (5%) | 787 (69%) | 872 (30%) |
| Missing | 3 (0%) | 0 (0%) | 3 (0%) |
| **Marital status** | | | |
| Unmarried | 885 (51%) | 229 (20%) | 1114 (39%) |
| Married/Cohabiting | 838 (48%) | 911 (80%) | 1749 (61%) |
| Missing | 5 (0%) | 0 (0%) | 5 (0%) |
| **Employment status** | | | |
| Unemployed | 985 (57%) | 270 (24%) | 1255 (44%) |
| Employed | 563 (33%) | 794 (70%) | 1357 (47%) |
| Missing | 180 (10%) | 76 (7%) | 256 (9%) |
| **Crowding index (persons/room)** | | | |
| Mean (SD) | 1.6 (± 0.86) | 0.74 (± 0.68) | 1.2 (± 0.89) |
| Missing | 14 (0.8%) | 16 (1.4%) | 30 (1.0%) |

*(Continued)*

**Table 1.** (Continued)

| Characteristics | Cape Town, South Africa (N = 1728) | Northern Plains, USA (N = 1140) | Overall (N = 2868) |
|---|---|---|---|
| **Maternal depression** | | | |
| None (EPDS score <13) | 846 (49%) | 1059 (93%) | 1905 (66%) |
| Probable depression (EPDS score 13–15) | 216 (12%) | 34 (3%) | 250 (9%) |
| Major depression (EPDS score >15) | 666 (39%) | 47 (4%) | 713 (25%) |
| **Prenatal drinking** | | | |
| None | 785 (45%) | 565 (50%) | 1350 (47%) |
| Moderate-high continuous | 325 (19%) | 63 (6%) | 388 (14%) |
| Low continuous | 86 (5%) | 13 (1%) | 99 (3%) |
| Quit early | 369 (21%) | 485 (43%) | 854 (30%) |
| Missing | 163 (9.4%) | 14 (1.2%) | 177 (6.2%) |
| **Prenatal smoking** | | | |
| None | 703 (41%) | 886 (78%) | 1589 (55%) |
| Moderate-high continuous | 392 (23%) | 48 (4%) | 440 (15%) |
| Low continuous | 573 (33%) | 124 (11%) | 697 (24%) |
| Quit early | 53 (3%) | 62 (5%) | 115 (4%) |
| Missing | 7 (0.4%) | 20 (1.8%) | 27 (0.9%) |
| **Pregnancy and delivery complications** | | | |
| **Gestational diabetes mellitus** | | | |
| No | 1635 (95%) | 977 (86%) | 2612 (91%) |
| Yes | 12 (1%) | 76 (7%) | 88 (3%) |
| Missing | 81 (4.7%) | 87 (7.6%) | 168 (5.9%) |
| **Gestational hypertension** | | | |
| No | 1604 (93%) | 1092 (96%) | 2696 (94%) |
| Yes | 69 (4%) | 41 (4%) | 110 (4%) |
| Missing | 55 (3.2%) | 7 (0.6%) | 62 (2.2%) |
| **Preeclampsia** | | | |
| No | 1617 (94%) | 1091 (96%) | 2708 (94%) |
| Yes | 56 (3%) | 42 (4%) | 98 (3%) |
| Missing | 55 (3.2%) | 7 (0.6%) | 62 (2.2%) |
| **Induced Labor** | | | |
| No | 1451 (84%) | 810 (71%) | 2261 (79%) |
| Yes | 205 (12%) | 315 (28%) | 520 (18%) |
| Missing | 72 (4.2%) | 15 (1.3%) | 87 (3.0%) |
| **Augmented labor** | | | |
| No | 1472 (85%) | 796 (70%) | 2268 (79%) |
| Yes | 184 (11%) | 329 (29%) | 513 (18%) |
| Missing | 72 (4.2%) | 15 (1.3%) | 87 (3.0%) |
| **Cesarean section** | | | |
| No | 1459 (84%) | 832 (73%) | 2291 (80%) |
| Yes | 261 (15%) | 289 (25%) | 550 (19%) |
| Missing | 8 (0.5%) | 19 (1.7%) | 27 (0.9%) |
| **Child factors** | | | |
| **Resuscitated at birth** | | | |
| No | 1587 (92%) | 907 (80%) | 2494 (87%) |
| Yes | 135 (8%) | 215 (19%) | 350 (12%) |

*(Continued)*

**Table 1.** (Continued)

| Characteristics | Cape Town, South Africa (N = 1728) | Northern Plains, USA (N = 1140) | Overall (N = 2868) |
|---|---|---|---|
| Missing | 6 (0.3%) | 18 (1.6%) | 24 (0.8%) |
| **NICU admission** | | | |
| No | 1515 (88%) | 980 (86%) | 2495 (87%) |
| Yes | 136 (8%) | 124 (11%) | 260 (9%) |
| Missing | 77 (4.5%) | 36 (3.2%) | 113 (3.9%) |
| **Gestational Age at Delivery (weeks)** | | | |
| Mean (SD) | 39 (± 2.1) | 39 (± 1.7) | 39 (± 2.0) |
| Missing | 0 (0%) | 4 (0.4%) | 4 (0.1%) |
| **Infant sex** | | | |
| Male | 844 (49%) | 563 (49%) | 1407 (49%) |
| Female | 884 (51%) | 573 (50%) | 1457 (51%) |
| Missing | 0 (0%) | 4 (0.4%) | 4 (0.1%) |
| **Infant birthweight** | | | |
| Very low birthweight (<2000g) | 70 (4%) | 19 (2%) | 89 (3%) |
| Low- moderate birthweight (2000-2500g) | 196 (11%) | 39 (3%) | 235 (8%) |
| Normal birthweight (>2500g) | 1454 (84%) | 1050 (92%) | 2504 (87%) |
| Missing | 8 (0.5%) | 32 (2.8%) | 40 (1.4%) |
| **Age of developmental assessment (months)** | | | |
| | 12.17 (0.40) | 12.39 (0.46) | 12.25 (0.44) |

## Predictors of child development in Northern Plains

The estimates of association of predictors with infant development in the Northern Plains site are shown in Figs 6–10. We found significant associations of high maternal BMI during pregnancy with cognitive and language scores. Compared to infants of mother with normal early pregnancy BMI (18.5−25 kg/m$^2$), infants of obese mothers had on average 0.21 SD (CI: −0.37, −0.06) lower cognitive, 0.18 SD (CI −0.32, −0.03) lower receptive language, and 0.13 SD (CI: −0.28, 0.00) lower expressive language scores on the MSEL. Low BMI (<18.5 kg/m$^2$) was also associated with 0.48 SD reduction in cognitive scores (CI: −0.90, −0.06). Among the indicators of socioeconomic status, crowding index was associated with cognitive and fine motor scores and maternal education was associated with fine motor scores. With each unit increase in the crowding index, there was 0.14 SD (CI: −0.26, −0.03) reduction in cognitive and a 0.10 SD (CI: −0.21, −0.02) reduction in fine motor scores. Infants of mothers with primary education had 0.97 SD (CI: −1.67, −0.26) lower fine motor scores compared to infants of mothers with more than high school education. We did not find any significant association of prenatal drinking, smoking, or maternal depression with development scores.

Similar to the CT site, being born with low birthweight and having admission to the NICU were both independently linked with lower developmental scores in the NP. Compared to infants with normal birthweight, infants born with very to moderate low birthweight (<2000g) and moderate low birthweight (2000−2500g) had −0.62 SD and −0.33 SD lower cognitive scores, respectively. Infants who were born with very to moderate low birthweight (−0.74 SD, CI: −1.28, −0.21) and moderate birthweight (−0.38 SD, −0.72, −0.05) had lower gross motor scores compared to normal birthweight infants. Infants who were admitted to NICU had 0.29 SD (95% CI: −0.50, −0.09) lower cognitive, 0.22 SD (95% CI: −0.41, −0.03)

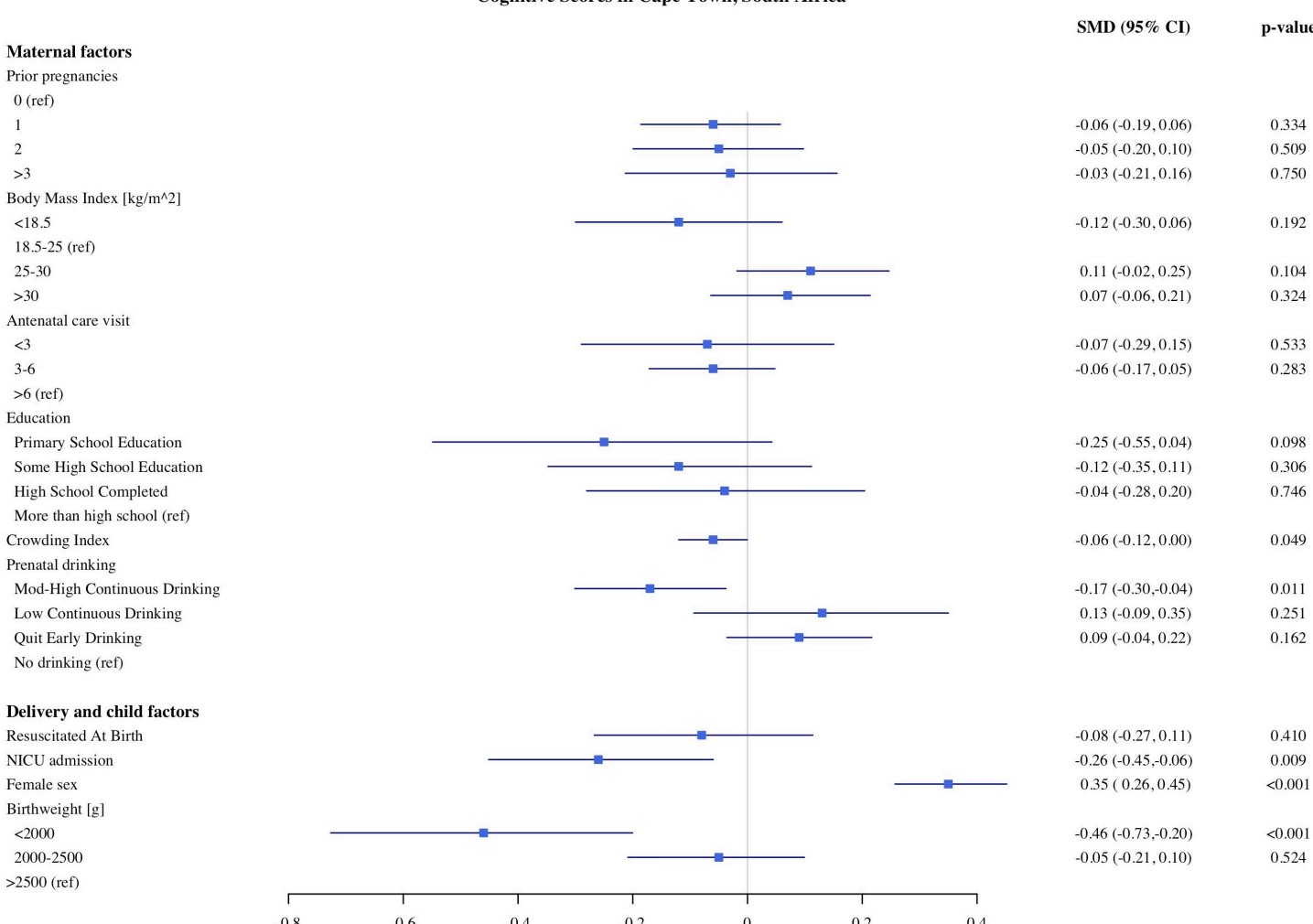

**Fig 1. Association of maternal, delivery and child factors with cognitive development in Cape Town, South Africa.**

lower receptive language scores and 0.33 SD (95% CI −0.54, −0.12) lower expressive language scores, compared to children who were not admitted to NICU. Across both sites, female infants had higher cognitive, and receptive and expressive language scores compared to male infants.

## Discussion

In this prospective cohort study among 2868 mother-infant dyads from the Northern Plains of the USA and Cape Town, South Africa, we identified associations of several prenatal maternal and birth factors with infant cognitive, language, and motor development at one year of age. We found a significant association of PAE with lower cognitive and language development of infants in Cape Town. Early pregnancy maternal obesity was associated with lower cognitive and language development scores in the Northern Plains. In addition, multiple risk factors classically associated with child development including lower levels of maternal educational attainment, crowded living conditions, being born with low birthweight, and NICU admission were significant predictors of lower neurodevelopmental scores in both study sites. We

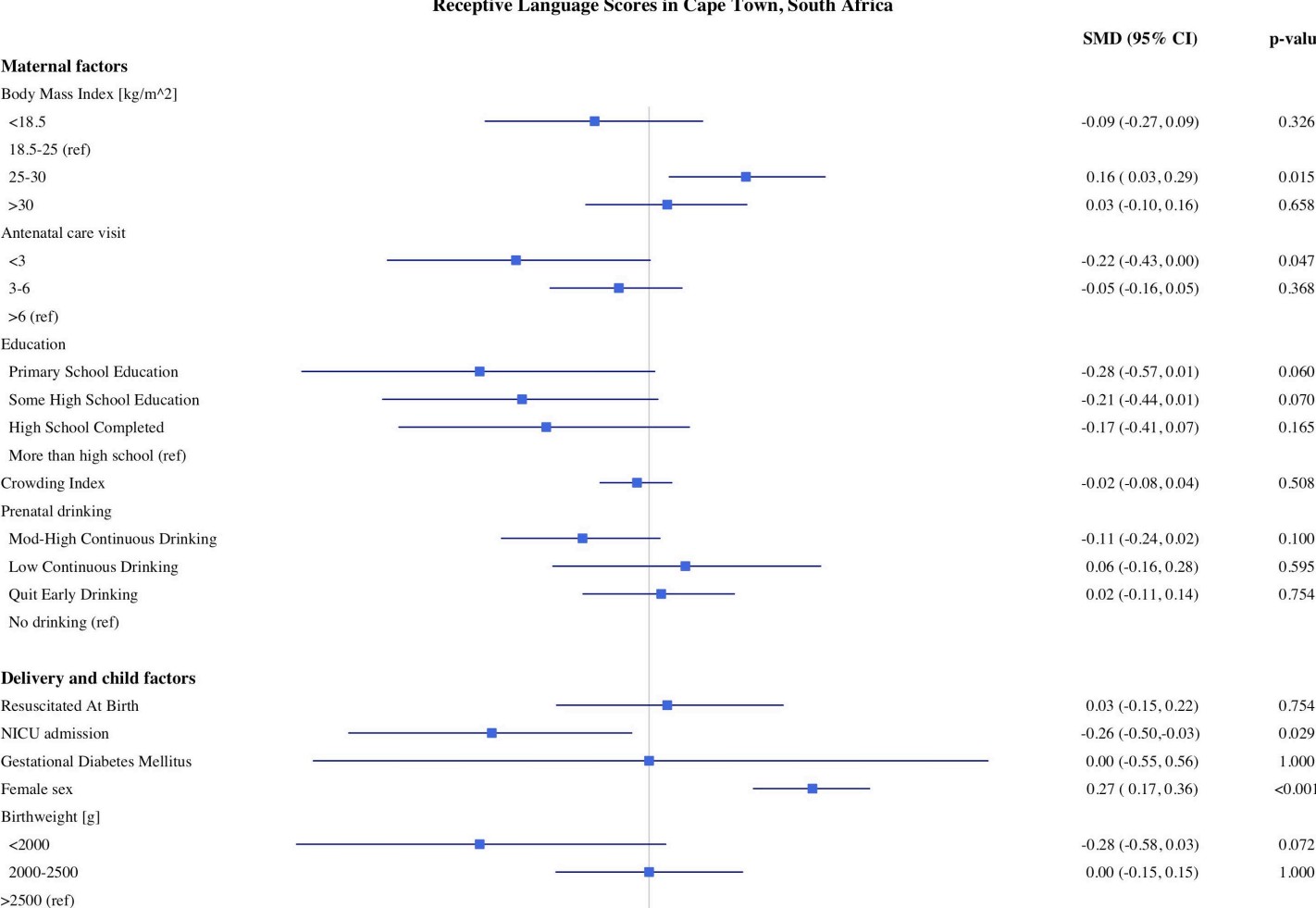

**Fig 2. Association of maternal, delivery and child factors with receptive language development in Cape Town, South Africa.**

did not find significant associations of prenatal tobacco use and pregnancy complications with infant development outcomes in these analyses.

We observed adverse effects of moderate to high continuous PAE on cognitive and language development in the Cape Town site. These findings align closely with previous studies from LMICs that have reported associations between prenatal drinking and deficits in attention, learning, and memory, as well as social and emotional development [40–43]. Similar to previous studies conducted in high income countries, we found no effects of PAE in the NP. One potential explanation is that the magnitude and duration of moderate-to-high-continuous PAE was much lower in the NP site (19% in CT compared to 4% in NP). This observation is supported by reports of dose-dependent effects of prenatal drinking, including a study in Danish population, which reported poor development among children whose mothers drank more than 9 drinks per week [44,45]. In addition, binge drinking patterns were more prevalent in the Cape Town site, which may have resulted in higher peak fetal alcohol exposure (S2 Table). The observed null effect in the NP site could also be, in part, due to a favorable postnatal environment. We accounted for the same set of uniformly measured confounding variables in both

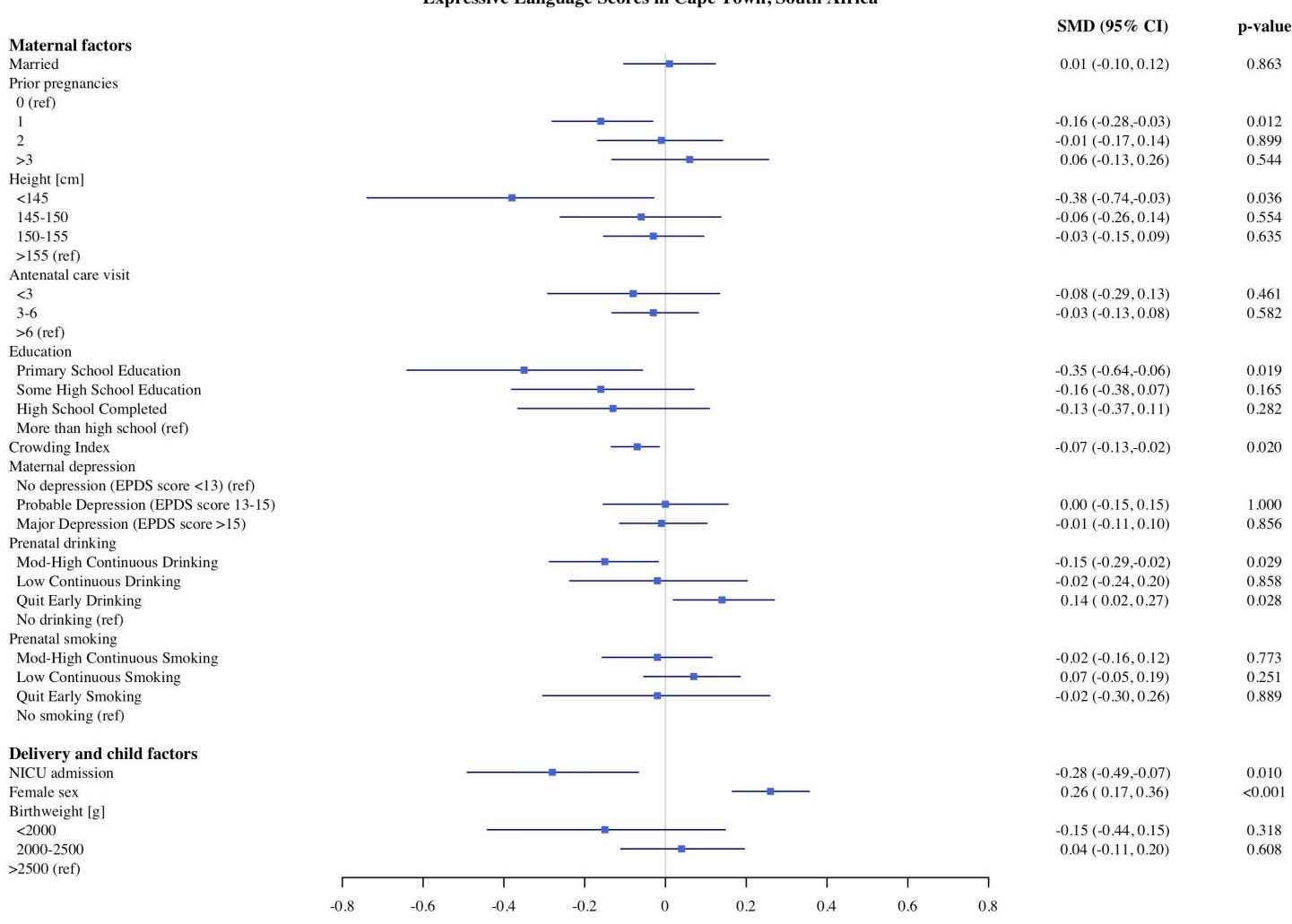

**Fig 3. Association of maternal, delivery and child factors with expressive language development in Cape Town, South Africa.**

study sites; therefore residual confounding is an unlikely explanation for the null effect in NP site. The differential impact of prenatal alcohol exposure (PAE) on development in South African infants, compared to American infants, can be due to the protective influence of adequate nutritional intake among populations in the United States, in contrast to the nutritional deficiencies prevalent in low-resource communities in South Africa [46,47]. Although alcohol is a teratogen that directly crosses the placenta into fetal blood circulation, prior research has demonstrated dietary deficiencies, such as the essential nutrient choline, can exacerbate the teratogenic effects of PAE on offspring physical and behavioral development [48–50].

Our findings indicating that maternal obesity is associated with lower cognitive and language development in the NP site are strengthened by research conducted in high income settings [51]. In addition to child cognition and language development, prior studies reported association of maternal obesity with increased risk for autism, ADHD, emotion regulation and behavioral problems in childhood [26,52]. Increased levels of prenatal maternal inflammation are

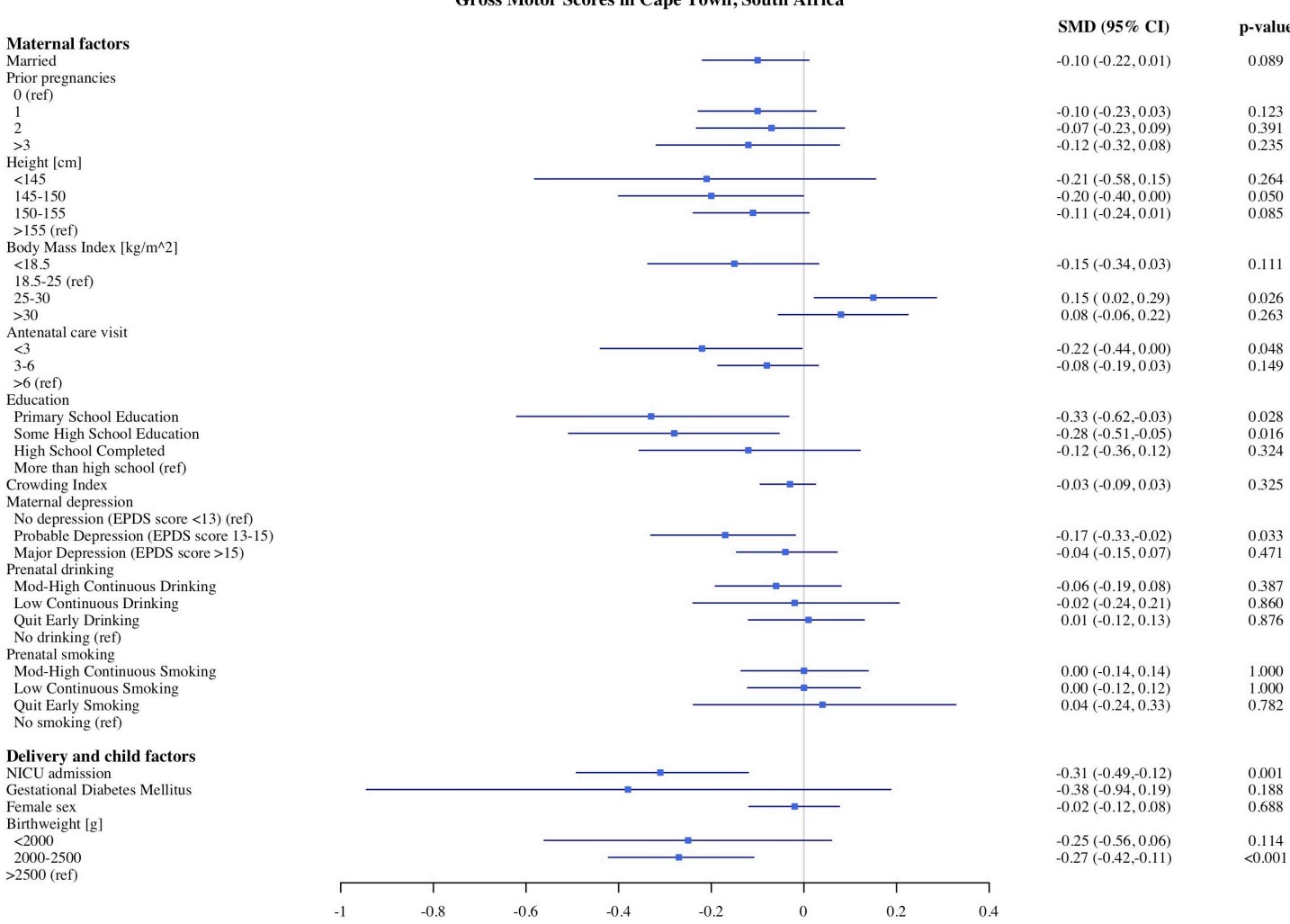

**Fig 4. Association of maternal, delivery and child factors with gross motor development in Cape Town, South Africa.**

a possible biological mechanism underlying these associations [53]. There is limited data on the effects of maternal obesity on child development from LMIC settings, although rates of maternal obesity are rising in those contexts [54]. Our observation of no association of maternal obesity in the Cape Town site could be due to lower rates of overweight and obesity (63% in NP compared to 42% in CT). In addition, BMI was measured later during pregnancy in the Cape Town site. BMI measured in later gestation, in part, reflects gestational weight gain. Thus, the null association in the CT site could also be explained by measurement error in the BMI assessment. Additionally, in low-resource settings where dietary inadequacy and micronutrient deficiencies are common, higher maternal BMI may not solely reflect excess caloric intake. In such contexts, overweight and obesity frequently coexist with micronutrient deficiencies among women of reproductive age, potentially influencing the relationship between maternal obesity and child developmental outcomes [55,56].

In our study, multiple indicators of socioeconomic status including maternal education and household crowding were associated with poor development in infants in both study sites, although the magnitude and domains of

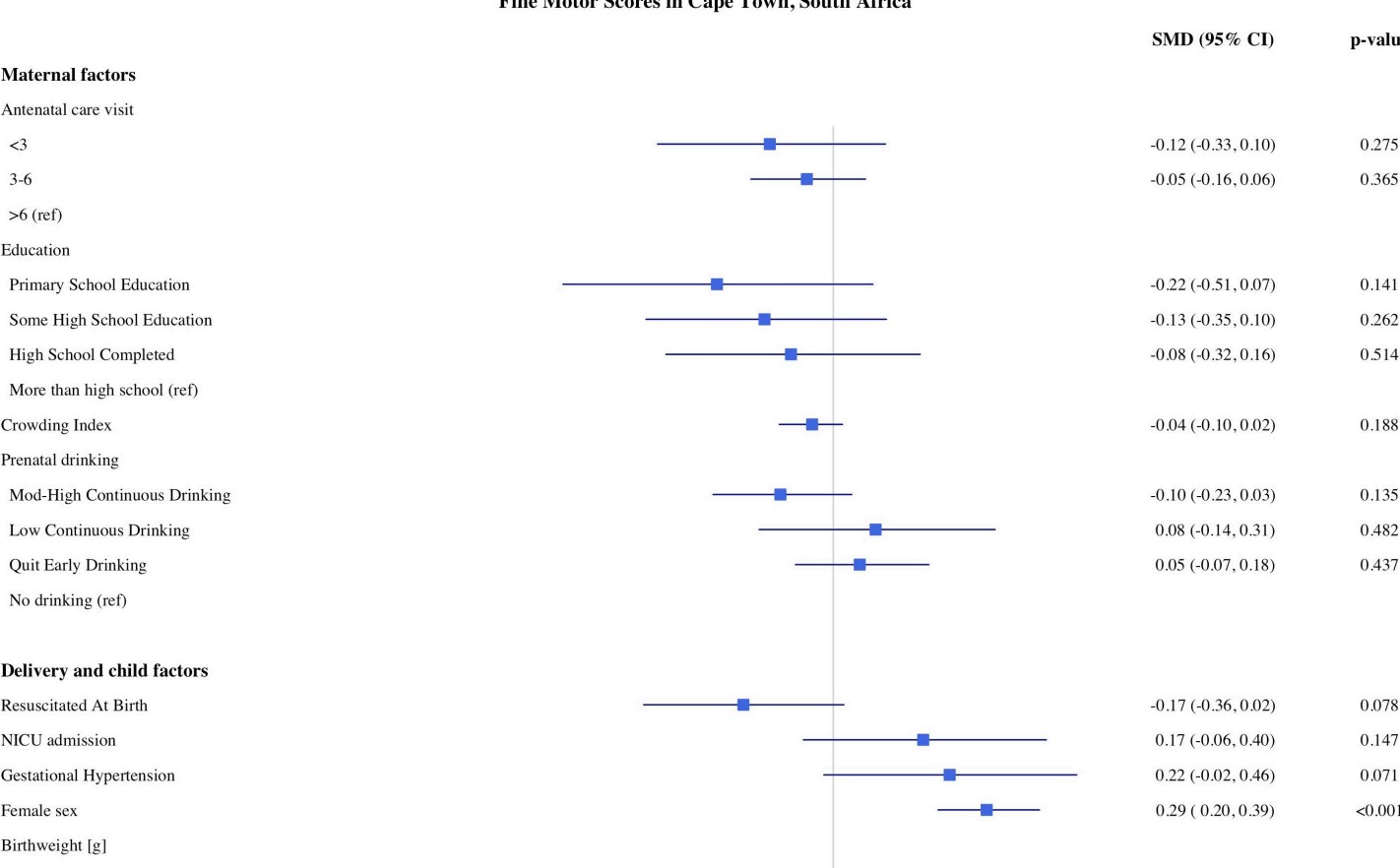

**Fig 5. Association of maternal, delivery and child factors with fine motor development in Cape Town, South Africa.**

association differed. The unique contribution of our study is that we demonstrate detrimental effects of household crowding on multiple domains of child development across study sites. Chronic residential crowding has been linked to behavior problems and lower educational attainment in high income settings [57,58]. Analysis of the cross-sectional survey data from LMICs shows association of household crowding with poor cognitive and socioemotional development [59]. The estimates of household crowding accounts for common correlates including parity and other indicators of SES, supporting that crowding impacts on infant development independently which is likely mediated via reduced maternal responsiveness in chaotic environments [60]. Taken together, our findings indicate that household crowding, along with other easily assessable markers including maternal educational attainment, low birthweight and NICU admissions, may help identify infants who could benefit from targeted developmental screening and early intervention.

The strength of our study includes prospective data from a large sample of infants from two socioeconomically and ethnically diverse populations. We collected data on prenatal exposures, socioeconomic status, nutritional status, and

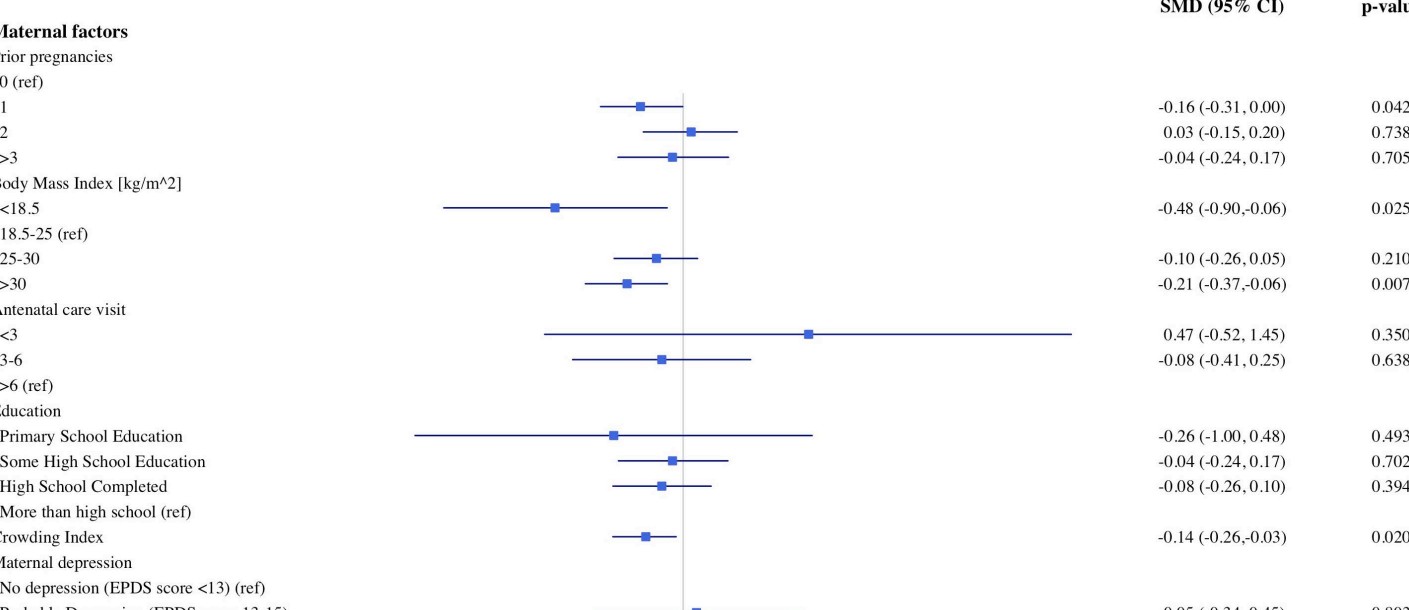

Cognitive Scores in the Northern Plains, USA

| | SMD (95% CI) | p-value |
|---|---|---|
| **Maternal factors** | | |
| Prior pregnancies | | |
| 0 (ref) | | |
| 1 | -0.16 (-0.31, 0.00) | 0.042 |
| 2 | 0.03 (-0.15, 0.20) | 0.738 |
| >3 | -0.04 (-0.24, 0.17) | 0.705 |
| Body Mass Index [kg/m^2] | | |
| <18.5 | -0.48 (-0.90,-0.06) | 0.025 |
| 18.5-25 (ref) | | |
| 25-30 | -0.10 (-0.26, 0.05) | 0.210 |
| >30 | -0.21 (-0.37,-0.06) | 0.007 |
| Antenatal care visit | | |
| <3 | 0.47 (-0.52, 1.45) | 0.350 |
| 3-6 | -0.08 (-0.41, 0.25) | 0.638 |
| >6 (ref) | | |
| Education | | |
| Primary School Education | -0.26 (-1.00, 0.48) | 0.493 |
| Some High School Education | -0.04 (-0.24, 0.17) | 0.702 |
| High School Completed | -0.08 (-0.26, 0.10) | 0.394 |
| More than high school (ref) | | |
| Crowding Index | -0.14 (-0.26,-0.03) | 0.020 |
| Maternal depression | | |
| No depression (EPDS score <13) (ref) | | |
| Probable Depression (EPDS score 13-15) | 0.05 (-0.34, 0.45) | 0.803 |
| Major Depression (EPDS score >15) | -0.17 (-0.51, 0.17) | 0.324 |
| | | |
| **Delivery and child factors** | | |
| Induced Labor | 0.05 (-0.10, 0.19) | 0.497 |
| Cesarean Section | -0.12 (-0.27, 0.03) | 0.109 |
| NICU admission | -0.29 (-0.50,-0.09) | 0.005 |
| Preeclampsia | -0.23 (-0.57, 0.11) | 0.180 |
| Gestational Diabetes Mellitus | -0.11 (-0.36, 0.13) | 0.376 |
| Female sex | 0.15 ( 0.02, 0.28) | 0.021 |
| Birthweight [g] | | |
| <2000 | -0.62 (-1.15,-0.10) | 0.021 |
| 2000-2500 | -0.33 (-0.67, 0.01) | 0.058 |
| >2500 (ref) | | |

**Fig 6. Association of maternal, delivery and child factors with cognitive development in Northern Plains, USA.**

other covariates using uniform data collection tools and assessment schedules which allowed comparability of the estimates across study sites. Our study has other limitations including lack of data on postnatal environmental exposures such as postnatal maternal depression, early stimulation and responsive parenting practices, or other environmental exposures which are major determinants of early childhood development [61]. The relatively limited duration of follow-up may have constrained our ability to detect exposure-related effects on neurodevelopmental outcomes that emerge later in childhood, when a broader range of cognitive and behavioral functions can be reliably assessed. We also did not adjust for multiple comparisons given the correlated nature of the outcomes and covariates and the a priori hypotheses guiding our analyses. Results should therefore be interpreted in the context of effect sizes, confidence intervals, and consistency across outcomes rather than statistical significance alone [37–39].Finally, we used data from the PASS study which was conducted among populations with high prevalence of prenatal drinking and smoking, therefore the results of this study may not be directly generalizable to populations with a low prevalence of these exposures.

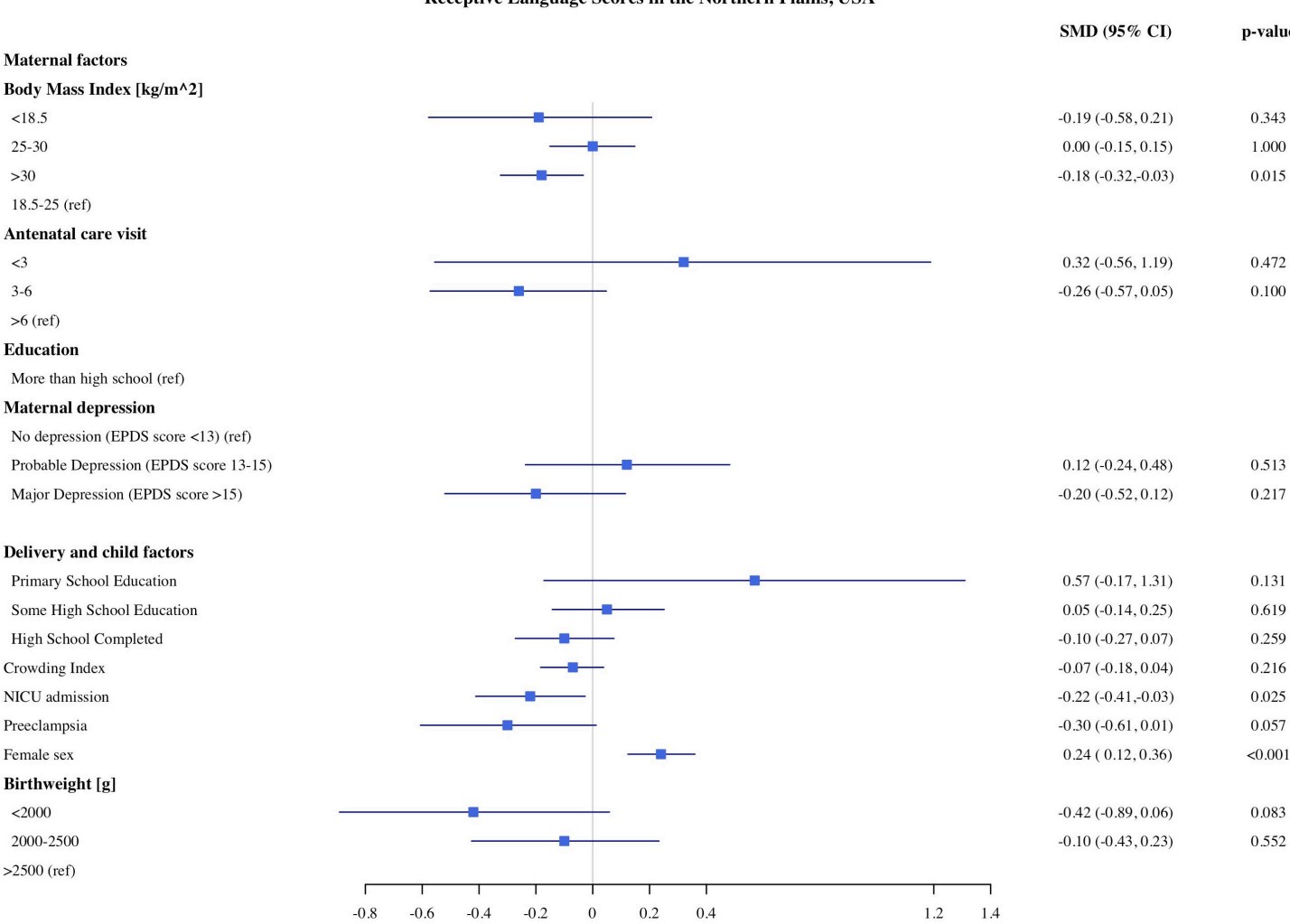

**Fig 7. Association of maternal, delivery and child factors with receptive language development in Northern Plains, USA.**

In summary, the contribution of our study is two-fold. First, we demonstrate the influence of risk factors classically associated with poor neurodevelopment is similar across study sites. Our findings indicate that household crowding, maternal educational attainment, low birthweight and NICU admission are easily assessable markers that can help identify infants in both high- and low-income settings for developmental screening and interventions. Second, our results support differential effects of PAE and maternal obesity across socioeconomic and cultural contexts. These findings highlight the importance of comprehensive intervention packages that are appropriate for the local context to be implemented for pregnant women and women of child-bearing age.

## Supporting information

**S1 Table. Amount smoking and drinking in the cluster groups by trimester.**
(DOCX)

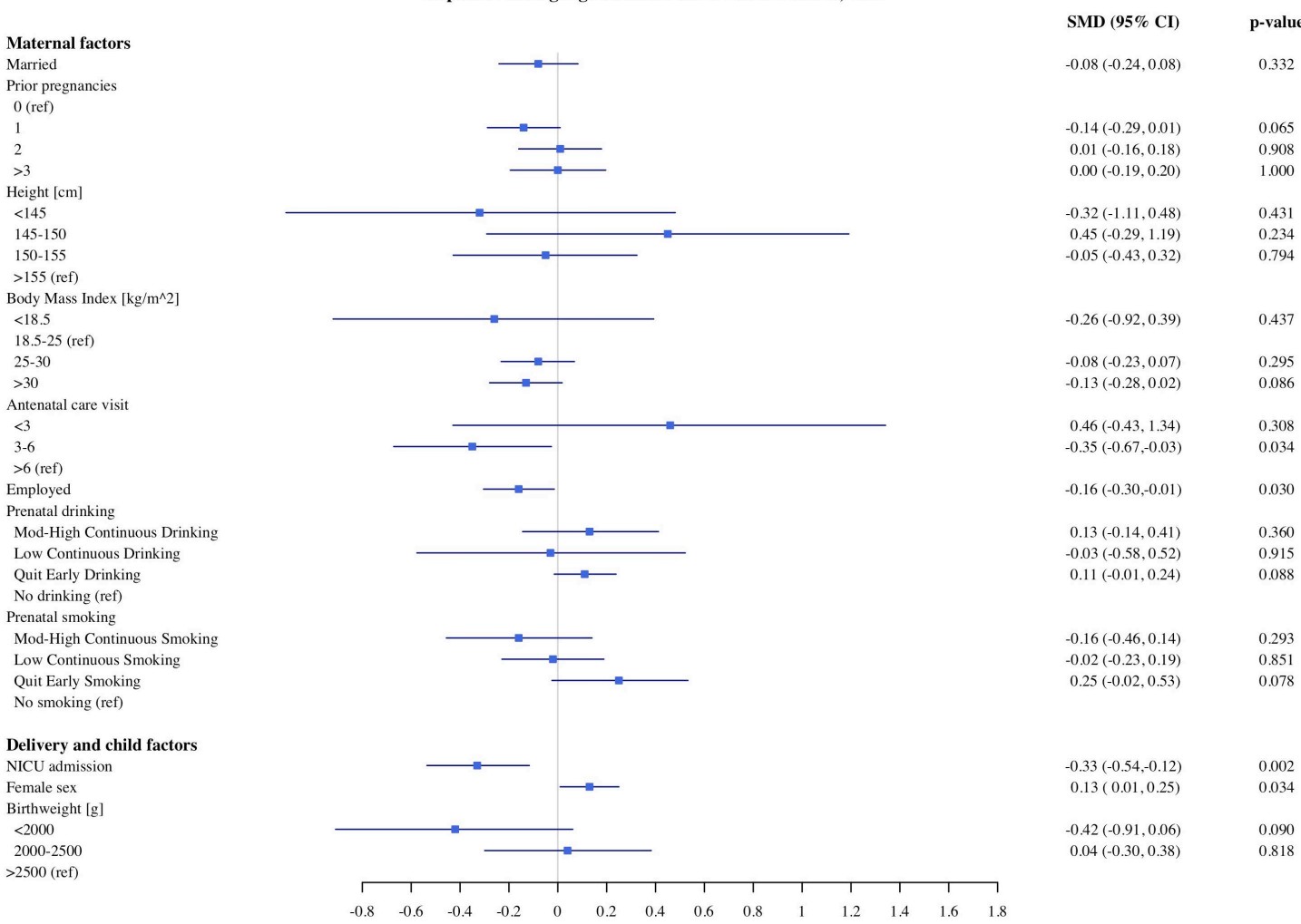

**Expressive Language Scores in the Northern Plains, USA**

| | SMD (95% CI) | p-value |
|---|---|---|
| **Maternal factors** | | |
| Married | -0.08 (-0.24, 0.08) | 0.332 |
| Prior pregnancies | | |
| 0 (ref) | | |
| 1 | -0.14 (-0.29, 0.01) | 0.065 |
| 2 | 0.01 (-0.16, 0.18) | 0.908 |
| >3 | 0.00 (-0.19, 0.20) | 1.000 |
| Height [cm] | | |
| <145 | -0.32 (-1.11, 0.48) | 0.431 |
| 145-150 | 0.45 (-0.29, 1.19) | 0.234 |
| 150-155 | -0.05 (-0.43, 0.32) | 0.794 |
| >155 (ref) | | |
| Body Mass Index [kg/m^2] | | |
| <18.5 | -0.26 (-0.92, 0.39) | 0.437 |
| 18.5-25 (ref) | | |
| 25-30 | -0.08 (-0.23, 0.07) | 0.295 |
| >30 | -0.13 (-0.28, 0.02) | 0.086 |
| Antenatal care visit | | |
| <3 | 0.46 (-0.43, 1.34) | 0.308 |
| 3-6 | -0.35 (-0.67, -0.03) | 0.034 |
| >6 (ref) | | |
| Employed | -0.16 (-0.30, -0.01) | 0.030 |
| Prenatal drinking | | |
| Mod-High Continuous Drinking | 0.13 (-0.14, 0.41) | 0.360 |
| Low Continuous Drinking | -0.03 (-0.58, 0.52) | 0.915 |
| Quit Early Drinking | 0.11 (-0.01, 0.24) | 0.088 |
| No drinking (ref) | | |
| Prenatal smoking | | |
| Mod-High Continuous Smoking | -0.16 (-0.46, 0.14) | 0.293 |
| Low Continuous Smoking | -0.02 (-0.23, 0.19) | 0.851 |
| Quit Early Smoking | 0.25 (-0.02, 0.53) | 0.078 |
| No smoking (ref) | | |
| **Delivery and child factors** | | |
| NICU admission | -0.33 (-0.54, -0.12) | 0.002 |
| Female sex | 0.13 (0.01, 0.25) | 0.034 |
| Birthweight [g] | | |
| <2000 | -0.42 (-0.91, 0.06) | 0.090 |
| 2000-2500 | 0.04 (-0.30, 0.38) | 0.818 |
| >2500 (ref) | | |

**Fig 8. Association of maternal, delivery and child factors with expressive language development in Northern Plains, USA.**

**S2 Table. Distribution of binge events by cluster group (Non-drinkers/Quit early/Low continuous/Moderate to high continuous), by site (Northern Plains, NP, Cape Town, CT), and trimester (T1/T2/T3).**
(DOCX)

**S1 Fig. Study flow diagram illustrating participant selection for the analytic sample.**
(JPG)

## Author contributions

**Conceptualization:** Ayesha Sania, Jyoti Angal, Hein Odendaal, Amy J. Elliott, William P. Fifer, Lauren C. Shuffrey.

**Data curation:** Ayesha Sania, Mandy Potter, Lucy Brink, Hein Odendaal, Amy J. Elliott.

**Formal analysis:** Ayesha Sania, Shreya Rao, Nicolò Pini, Yael Rayport, Liana Eisler, Hein Odendaal, Amy J. Elliott, Lauren C. Shuffrey.

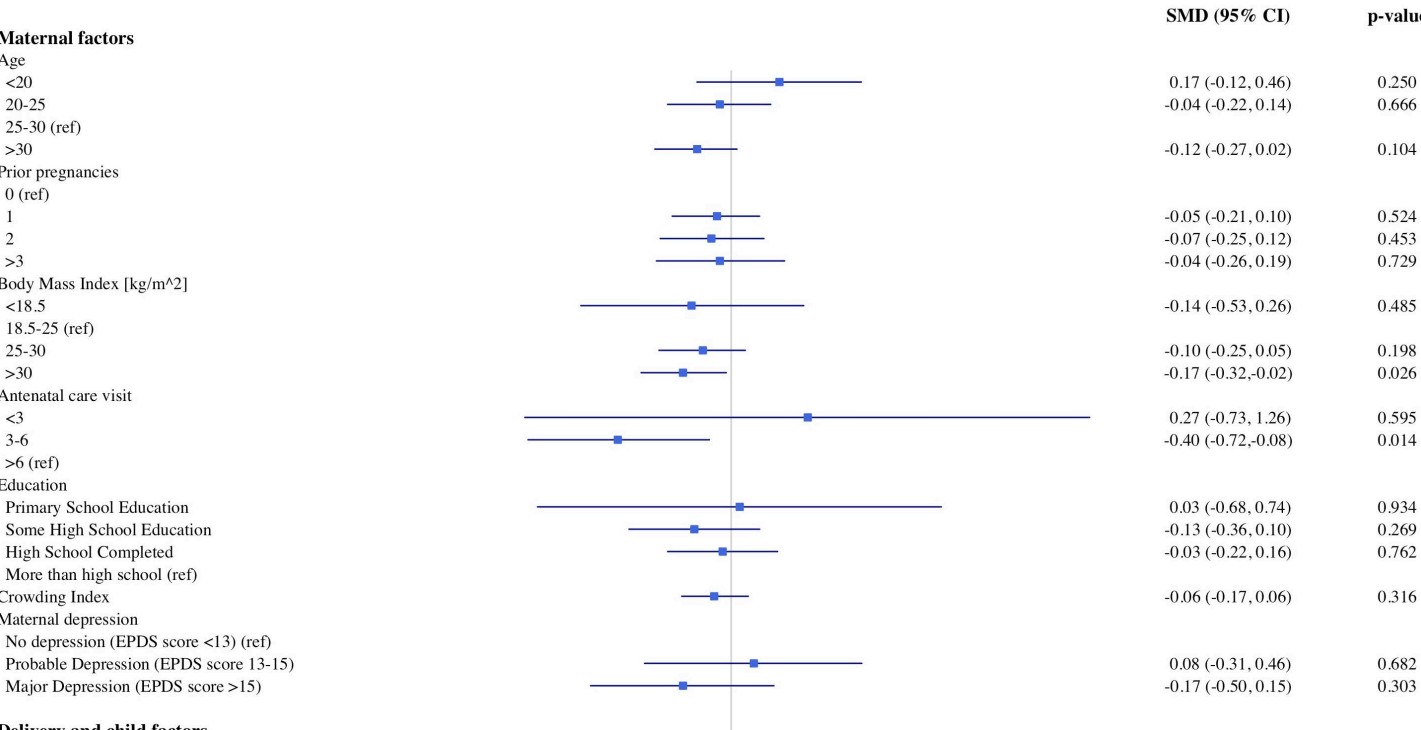

**Fig 9. Association of maternal, delivery and child factors with gross motor development in Northern Plains, USA.**

**Funding acquisition:** Hein Odendaal, Amy J. Elliott, William P. Fifer.

**Investigation:** Ayesha Sania, Liana Eisler, William P. Fifer, Lauren C. Shuffrey.

**Methodology:** Lucy Brink, Michael M. Myers.

**Project administration:** Mandy Potter, Lucy Brink, Jyoti Angal, Hein Odendaal, Amy J. Elliott.

**Supervision:** William P. Fifer.

**Visualization:** Ayesha Sania, Nicolò Pini, Michael M. Myers.

**Writing – original draft:** Ayesha Sania, Jyoti Angal, Lauren C. Shuffrey.

**Writing – review & editing:** Ayesha Sania, Shreya Rao, Nicolò Pini, Mandy Potter, Yael Rayport, Liana Eisler, Lucy Brink, Jyoti Angal, Michael M. Myers, Hein Odendaal, Amy J. Elliott, William P. Fifer, Lauren C. Shuffrey.

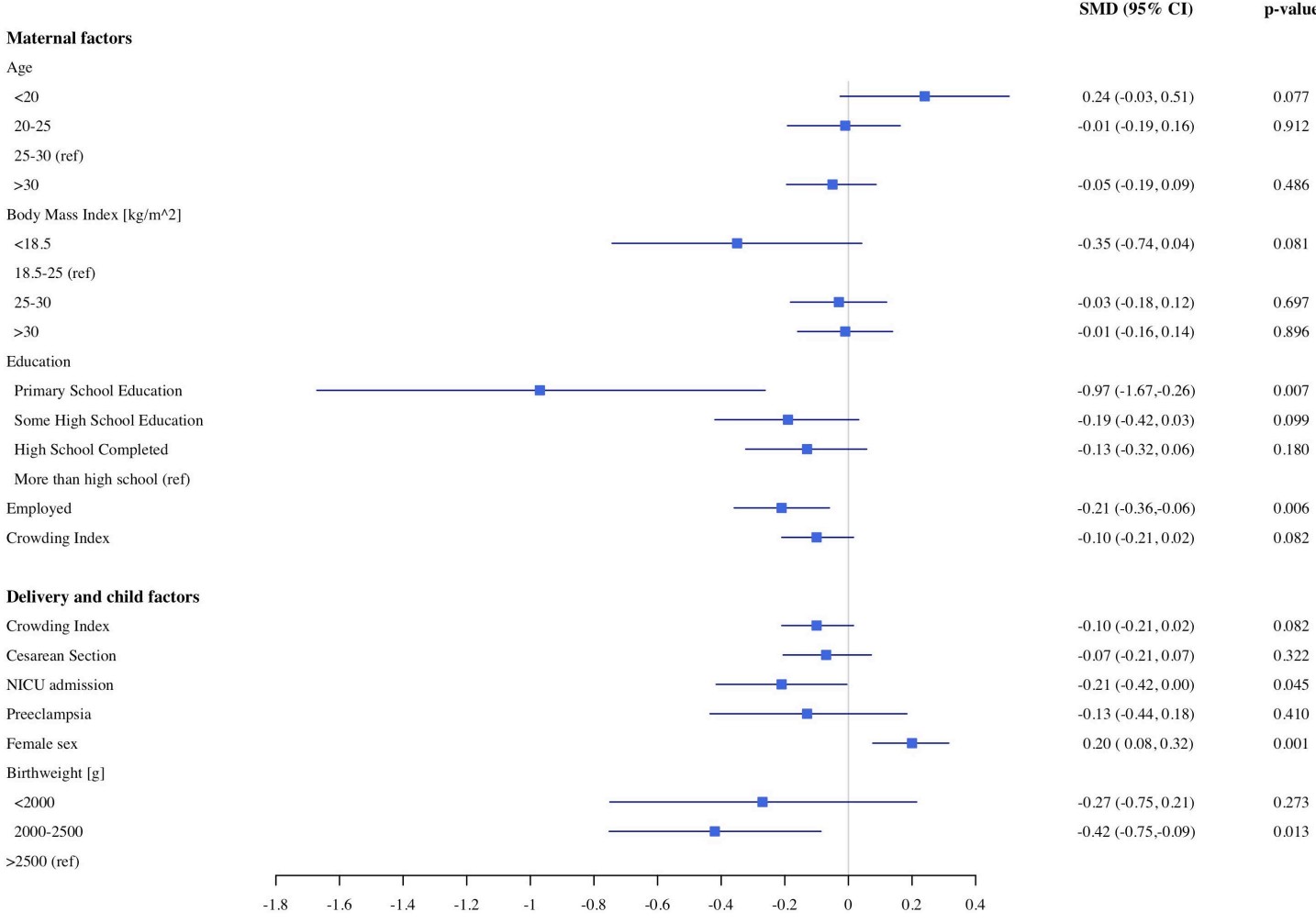

**Fig 10. Association of maternal, delivery and child factors with fine motor development in Northern Plains, USA.**

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
