## [Decision Letter · Decision Letter 0]

3 Sep 2025

PONE-D-25-11614

Maternal obesity and prenatal alcohol exposure are associated with child development: Results from the Safe Passage Study

PLOS ONE

Dear Dr. Sania,

Thank you for submitting your manuscript to PLOS ONE. After careful consideration, we feel that it has merit but does not fully meet PLOS ONE’s publication criteria as it currently stands. Therefore, we invite you to submit a revised version of the manuscript that addresses the points raised during the review process.

We look forward to receiving your revised manuscript.

Kind regards,

Emma K. Kalk

Academic Editor

PLOS ONE

Journal Requirements:

“This research was supported by grants UH3OD023279, U01HD055154, U01HD045935, U01HD055155, and U01AA016501, issued by the Office of the Director of the National Institutes of Health, National Institute on Alcohol Abuse and Alcoholism, Eunice Kennedy Shriver National Institute of Child Health and Human Development, and the National Institute on Deafness and Other Communication Disorders. Ayesha Sania was supported by a career development award from the Fogarty International Center at the NIH (1K01TW012425-01A1). The opinions expressed in this paper are those of the authors and do not necessarily represent the official views of the National Institutes of Health, the Eunice Kennedy Shriver National Institute of Child Health and Development, or Fogarty International Center, the National Institute on Alcohol Abuse and Alcoholism, or the National Institute on Deafness and Other Communication Disorders.”

Additional Editor Comments:

**Many thanks for your submission. Please will you address the Reviewers' Comments below and in the attached PDF.**

**Apologies for the delay.**

Reviewer's Responses to Questions

**Comments to the Author**

1. Is the manuscript technically sound, and do the data support the conclusions?

Reviewer #1: Yes

Reviewer #2: Yes

2. Has the statistical analysis been performed appropriately and rigorously?

Reviewer #1: Yes

Reviewer #2: Yes

3. Have the authors made all data underlying the findings in their manuscript fully available?

Reviewer #1: Yes

Reviewer #2: No

4. Is the manuscript presented in an intelligible fashion and written in standard English?

Reviewer #1: Yes

Reviewer #2: Yes

5. Review Comments to the Author

Reviewer #1: The study aimed to evaluate cognitive, motor, and language development of 1-year old children by using the Mullen Scales of Early Learning in a prospective cohort study of mother-infant dyads from Northern Plains (NP), USA and Cape Town (CT), South Africa. The relative contributions of prenatal alcohol use, maternal socioeconomic, and nutritional status were investigated within each study site.

The study presents the results of original research. Strengths include the prospective design, the large study sample, inclusion of both high- and low-income settings and measurements by trained assessors using standardized methods. Limitations include assessors not blinded (?) to predictor variables and the short duration of follow-up. I think these should be addressed in the Discussion section.

I have some detailed comments that follow below:

ABSTRACT

1. In the abstract, data from Northern Plains (NP) are presented first, however results starts with Cape Town (CT). I would consistently refer to CT first, then NP – throughout the abstract and manuscript

2. In the abstract, only prenatal alcohol use, maternal socioeconomic and nutritional status are mentioned in the first paragraph, while prenatal and delivery factors are mentioned in the second paragraph, perhaps mention the three main statistical models, i.e. maternal (or prenatal) factors, pregnancy complications and child factors?

INTRODUCTION

1. I was surprised about the potential positive effects of alcohol on cognition in high income settings, and I think the phrasing of the sentence could be misleading, even though the authors explain that it may be an effect of socioeconomic status. There are other references stating the harmful effects of alcohol also in high-income settings, for instance:

a. Hoyme HE, Kalberg WO, Elliott AJ, Blankenship J, Buckley D, Marais AS et al. Updated clinical guidelines for diagnosing fetal Alcohol Spectrum disorders. Pediatrics. 2016;138(2).

b. Lange S, Probst C, Gmel G, Rehm J, Burd L, Popova S. Global prevalence of fetal alcohol spectrum disorder among children and youth: a systematic review and Meta-analysis. JAMA Pediatr. 2017;171(10):948–56.

c. Jacobson JL, Akkaya-Hocagil T, Ryan LM, Dodge NC, Richardson GA, Olson HC, et al. Effects of prenatal alcohol exposure on cognitive and behavioral development: findings from a hierarchical meta-analysis of data from six prospective longitudinal U.S. cohorts. Alcohol Clin Exp Res. 2021;45(10):2040–58

METHODS

2. Page 7, line 173-175: Are maternal socioeconomic indicators included in both Model 1 and 2?

3. Where the trained assessors performing the Mullen blinded to the predictive factors? That would be a strength that could be emphasized. In case they were not blinded this potential bias should be discussed in Discussion

RESULTS

4. I would be consistent throughout the manuscript starting with the CT results

5. Page 8, line 198-199: Perhaps use the abbreviations CT and NP once they are introduced?

6. I was also curious to know which factors contributed the most to developmental outcomes, the mother or the child factors – perhaps this could be discussed in the Discussion

DISCUSSION

7. Also here I would be consistent starting with CT

8. I would address limitations such as assessors not being blinded to predictors (if that was the case) and the limited follow-up period of 1 year – as children are in continuous development, and 1 year is too early to determine any long-term effects

FIGURES AND TABLES

9. Is there a typo in Table 1: Northern Plaines? (should be Northern Plains)

Reviewer #2: SUMMARY:

This is a secondary analysis of a large cohort study of mother-infant dyads in two areas with high rates of maternal alcohol use. It assesses infant developmental outcomes at 1 year of age in relation to a variety of demographic, prenatal and perinatal exposures using multiple logistic regression analysis controlling for various confounding variables in 3 categories. A number of known demographic and perinatal risk factors for poor developmental outcomes were confirmed to be present in both cohorts. There were many differences in exposures between the Cape Town and Northern Plains sites, though the sites were not compared statistically, with the analysis being conducted internally for each site. For example prenatal alcohol and smoking exposures were much more common in the Cape Town site, and obesity more common in the Northern Plains site. High and ongoing prenatal alcohol exposure with associated with poor developmental outcome in the CT site but not the NP site, whereas the reverse was true for maternal obesity. Prenatal smoking exposure was not associated with poor developmental outcome. The authors concluded that risk factors could potentially be used to identify infants for early developmental interventions, but that these should context specific given the differences found between sites.

MAJOR COMMENTS:

The study is a well designed large cohort study. The cohort, studies methods and primary outcomes have been described in other articles. The findings presented in this paper do not appear to have been published elsewhere and are interesting in themselves. It is stated (line 106) that a subset of women enrolled before 24 weeks were randomly selected for further procedures. Please indicate briefly the process of selection.

At the beginning of the introduction it is mentioned that early cognitive... development is a key determinant of future educational and income achievement (line 54-55). It is not discussed how best to measure this and it should be discussed to what extent developmental outcomes at 1 year are sufficient for this purpose.

The introduction mentions that PAE is more risky in low income settings (line 69) than high income settings and various potential protective factors in high income settings are mentioned. The role of additional risk factors in low income settings should also be mentioned e.g. binge drinking is known to be a common pattern in South Africa.

In the methods section on Study Populations (line 96-109), little background is given regarding the nature of the two study sites. Some further context would help readers to interpret the article e.g. are the residential areas urban vs rural, how do SES generally compare to average in each country?

3750 infants were selected for further procedures (line 106), and 2868 (line 193) were included in this study (i.e. 24% dropouts). Please specify the reasons for dropouts per study site.

The manner in which alcohol and smoking exposures were stratified was described in a previous paper by Dukes et al (2017) and is statistically sophisticated. It allows for assessment of longitudinal drinking and smoking practices through pregnancy, but makes it less obvious to the reader how risky is the drinking at any particular timepoint. Details of drinking and smoking are describe for the cohort as a whole in supplementary table 1. It would be helpful to include rates of binge drinking per trimester, since binge drinking is a specific risk for adverse outcomes of PAE. This could be included in Supplementary Table 1 and/or text, and does not require additional statistical analysis. Given that the analyses are done for the two sites independently, Supplementary Table 1 should give drinking and smoking for the two sites separately. Also, given that substance exposures are a particular point of interest for the Safe Passage study it should be stated why other drug use is not included in the analysis.

On lines 145-6, it is stated that the standard number of drinks of alcohol consumed per day of pregnancy was estimated. This is expressed in Supplementary table 1 as the drinks/trimester. Therefore perhaps better to state here also drinks/trimester rather than drinks/day.

Figures 1-10 outline neurodevelopmental outcomes at each site by risk factor. These are presented as SMDs with 95% CIs. P-values are not included, and should be included. The risk factors included vary considerably between figures - please indicate in the text why this is the case and try to standardize as much as possible which variables are included, and try to at least include smoking in all figures given that it is a stated point of interest.

It is stated in the text (line 217) that in CT probable depression was associated with lower gross motor scores than no depression. It think this should be tempered with a statement that there was a big group with major depression and this was not associated with lower gross motor score (so there is not apparent "dose-response" effect).

In the discussion, it is mentioned that multiple factors classically associated with child development showed similar effects across sites (lines 269-272). Child sex should be included on the list (as an aside, the effect of child sex seems particular large in the CT site, though this is probably hard to interpret). You end up proposing variables to identify infants for early intervention - should child sex be one of these?

In relation to the fact that PAE affected developmental outcome in CT but not NP you indicate (line 287) that residual confounding is unlikely due to a uniform set of confounding variables being assessed. Is it not possible that unmeasured confounder/s could have affected results? (e.g. drug use, for argument sake). In addition, there is evidence that the blood alcohol concentration is an important determinant of risk related to PAE - BAC is determined not just by the level of alcohol consumption but also by volume of distribution/body size (women in NP were bigger - likely reducing effect of PAE), and drinking behavior that was not described (e.g. binges). It would therefore be appropriate to add some further nuance to the discussion of dietary deficiencies (line 289-295).

Maternal obesity was associated with developmental outcome in NP but not CT. It is suggest that this may be due to lower levels of obesity in CT or due to measurement error in CT due to maternal weight being measured on average at 20 weeks vs 16 weeks (average gestations at enrollment) (lines 304-308). I find this unconvincing - a considerable number of individuals in CT measured as obese, and the 4 week difference does not feel obviously sufficient to mask an effect. It seems equally plausible that other explanations are possible (though speculative) e.g. in an environment where dietary deficiencies are more common obesity may be associated with less deficiency of specific nutrients.

The fact that this is a secondary analysis means there should be some caution about the results. This is especially true given the large number of statistical comparisons and the lack of statistical adjustment for this. For example it is not obvious why pre-eclampsia would be associated with a better receptive language score in CT (figure 2), or why resuscitation at birth would be associated with better outcome in NP (figure 7) and this result could perhaps be the result of multiple comparisons? The role of multiple comparisons should be noted in the limitations, which should be stated in more detail. The use of developmental outcomes at 1 year is an imperfect proxy for long term development (outcome at 3 or 5 years of age would be better) - this is also a limitation to be mentioned.

MINOR:

A few typographical errors are highlighted in the uploaded pdf.

6. PLOS authors have the option to publish the peer review history of their article (what does this mean?). If published, this will include your full peer review and any attached files.

Reviewer #1: No

Reviewer #2: No

---

## [Author Response · Author response to Decision Letter 1]

27 Feb 2026

We are thankful to the editor and to the reviewers for the thoughtful comments of our manuscript. We have provided both the highlighted version and a clean version of the revised manuscript. As requested, below, we have provided a response to each editor and reviewer comment (indicated in italics) and identified the page and paragraph numbers where the changes (indicated in bold) have been made to the clean revised manuscript. Additionally, minor edits were made to the manuscript for consistency, clarification and simplification and are noted in the highlighted version of the revised manuscript.

Editor:

Response: We have reviewed the guidelines and formatted the manuscript accordingly.

“This research was supported by grants UH3OD023279, U01HD055154, U01HD045935, U01HD055155, and U01AA016501, issued by the Office of the Director of the National Institutes of Health, National Institute on Alcohol Abuse and Alcoholism, Eunice Kennedy Shriver National Institute of Child Health and Human Development, and the National Institute on Deafness and Other Communication Disorders. Ayesha Sania was supported by a career development award from the Fogarty International Center at the NIH (1K01TW012425-01A1). The opinions expressed in this paper are those of the authors and do not necessarily represent the official views of the National Institutes of Health, the Eunice Kennedy Shriver National Institute of Child Health and Development, or Fogarty International Center, the National Institute on Alcohol Abuse and Alcoholism, or the National Institute on Deafness and Other Communication Disorders.”

Response: We have added this statement to the cover letter.

3. We note that you have indicated that there are restrictions to data sharing for this study. PLOS only allows data to be available upon request if there are legal or ethical restrictions on sharing data publicly. For more information on unacceptable data access restrictions, please see http://journals.plos.org/plosone/s/data-availability#loc-unacceptable-data-access-restrictions. Before we proceed with your manuscript, please address the following prompts:

https://journals.plos.org/plosone/s/recommended-repositories. You also have the option of uploading the data as Supporting Information files, but we would recommend depositing data directly to a data repository if possible. We will update your Data Availability statement on your behalf to reflect the information you provide.

Response: We have updated the data availability statement in the online submission system and in the cover letter. We now provide a location where data can be accessed.

Data Availability: De-identified data from the Safe Passage Study is available through NICHD’s Data and Specimen Hub (DASH). All cases of demographic and exposure data are available on DASH. Elliott, Amy (2025). A Prospective Study on the Role of Prenatal Alcohol Exposure in SIDS and Stillbirth (Version 1). NICHD Data and Specimen Hub. https://doi.org/10.57982/sv8c-4y07. The tribal data used in this study is restricted access per the requirements of participating tribal nations and the Indian Health Service IRB. Avera Health maintains the data on a secure server, and people can contact Dr. Christine Hockett (Christine.hockett@avera.org) to learn the process for gaining tribal approval and the necessary regulatory approvals to gain access.

Response: We now include full names of the IRB in the methods section (Page 8, Paragraph 2)

Response: Thank you for this guidance. We have reviewed the articles proposed by reviewers and included the relevant ones.

Response: We have reviewed the reference list and updated to PLOS citation style

Reviewer #1:

The study aimed to evaluate cognitive, motor, and language development of 1-year old children by using the Mullen Scales of Early Learning in a prospective cohort study of mother-infant dyads from Northern Plains (NP), USA and Cape Town (CT), South Africa. The relative contributions of prenatal alcohol use, maternal socioeconomic, and nutritional status were investigated within each study site.

The study presents the results of original research. Strengths include the prospective design, the large study sample, inclusion of both high- and low-income settings and measurements by trained assessors using standardized methods. Limitations include assessors not blinded (?) to predictor variables and the short duration of follow-up. I think these should be addressed in the Discussion section.

I have some detailed comments that follow below:

ABSTRACT

1. In the abstract, data from Northern Plains (NP) are presented first, however results starts with Cape Town (CT). I would consistently refer to CT first, then NP – throughout the abstract and manuscript

Response: Thank you for pointing out this. We have revised the abstract and throughout the manuscript to present results from Cape Town first.

2. In the abstract, only prenatal alcohol use, maternal socioeconomic and nutritional status are mentioned in the first paragraph, while prenatal and delivery factors are mentioned in the second paragraph, perhaps mention the three main statistical models, i.e. maternal (or prenatal) factors, pregnancy complications and child factors?

Response: We have revised the abstract. We now mention the three models in the methods part of the abstract (Page 2).

INTRODUCTION

1. I was surprised about the potential positive effects of alcohol on cognition in high income settings, and I think the phrasing of the sentence could be misleading, even though the authors explain that it may be an effect of socioeconomic status. There are other references stating the harmful effects of alcohol also in high-income settings, for instance:

a. Hoyme HE, Kalberg WO, Elliott AJ, Blankenship J, Buckley D, Marais AS et al. Updated clinical guidelines for diagnosing fetal Alcohol Spectrum disorders. Pediatrics. 2016;138(2).

b. Lange S, Probst C, Gmel G, Rehm J, Burd L, Popova S. Global prevalence of fetal alcohol spectrum disorder among children and youth: a systematic review and Meta-analysis. JAMA Pediatr. 2017;171(10):948–56.

c. Jacobson JL, Akkaya-Hocagil T, Ryan LM, Dodge NC, Richardson GA, Olson HC, et al. Effects of prenatal alcohol exposure on cognitive and behavioral development: findings from a hierarchical meta-analysis of data from six prospective longitudinal U.S. cohorts. Alcohol Clin Exp Res. 2021;45(10):2040–58

Response: We have added a statement about harmful effects of drinking in high income settings and cited the Jacobson et al and Lange et al papers. (Page 3, Paragraph 2)

METHODS

2. Page 7, line 173-175: Are maternal socioeconomic indicators included in both Model 1 and 2?

Response: We clarified that all models include maternal socioeconomic condition. (Page 8, Paragraph 1)

3. Were the trained assessors performing the Mullen blinded to the predictive factors? That would be a strength that could be emphasized. In case they were not blinded this potential bias should be discussed in Discussion.

Response: We now mention in the method section that the assessors were blinded of the exposure status. (Page 6, Paragraph 1)

RESULTS

4. I would be consistent throughout the manuscript starting with the CT results

Response: Thank you for this helpful suggestion. We have revised the Results section to consistently present findings from the Cape Town site first throughout the manuscript. In the Discussion, results are organized thematically to support interpretation and synthesis across sites rather than strictly by site order.

5. Page 8, line 198-199: Perhaps use the abbreviations CT and NP once they are introduced?

Response: Thank you. We have made these changes.

6. I was also curious to know which factors contributed the most to developmental outcomes, the mother or the child factors – perhaps this could be discussed in the Discussion

Response: We appreciate this thoughtful suggestion. Our analyses were designed to estimate associations between individual prenatal and contextual risk factors and developmental outcomes, rather than to compare their relative population-level contributions. Direct comparison of effect sizes across maternal and child factors would require consideration of exposure prevalence and potentially estimation of population attributable risk, which was beyond the scope of the present study.

DISCUSSION

7. Also here I would be consistent starting with CT

Response: We have revised the Discussion to present Cape Town findings first. In instances where cross-site comparisons are discussed thematically, we structured the text to optimize interpretability and conceptual flow rather than adhere strictly to site order.

8. I would address limitations such as assessors not being blinded to predictors (if that was the case) and the limited follow-up period of 1 year – as children are in continuous development, and 1 year is too early to determine any long-term effects

Response: Thank you for this important comment. We have revised the Discussion to explicitly acknowledge the limited duration of follow-up and clarify that developmental outcomes at 1 year represent early indicators rather than long-term effects. (Page 17, Paragraph 2).

FIGURES AND TABLES

9. Is there a typo in Table 1: Northern Plaines? (should be Northern Plains)

Response: We have corrected this typo. Thank you.

Reviewer #2:

SUMMARY:

This is a secondary analysis of a large cohort study of mother-infant dyads in two areas with high rates of maternal alcohol use. It assesses infant developmental outcomes at 1 year of age in relation to a variety of demographic, prenatal and perinatal exposures using multiple logistic regression analysis controlling for various confounding variables in 3 categories. A number of known demographic and perinatal risk factors for poor developmental outcomes were confirmed to be present in both cohorts. There were many differences in exposures between the Cape Town and Northern Plains sites, though the sites were not compared statistically, with the analysis being conducted internally for each site. For example prenatal alcohol and smoking exposures were much more common in the Cape Town site, and obesity more common in the Northern Plains site. High and ongoing prenatal alcohol exposure with associated with poor developmental outcome in the CT site but not the NP site, whereas the reverse was true for maternal obesity. Prenatal smoking exposure was not associated with poor developmental outcome. The authors concluded that risk factors could potentially be used to identify infants for early developmental interventions, but that these should context specific given the differences found between sites.

MAJOR COMMENTS:

1. The study is a well designed large cohort study. The cohort, studies methods and primary outcomes have been described in other articles. The findings presented in this paper do not appear to have been published elsewhere and are interesting in themselves. It is stated (line 106) that a subset of women enrolled before 24 weeks were randomly selected for further procedures. Please indicate briefly the process of selection.

Response: Thank you for this comment. We have clarified the random selection process in the Methods section, including the timing and eligibility criteria for inclusion in the embedded study. Specifically, approximately one in three eligible and consenting women enrolled before 24 weeks’ gestation was randomly selected at the recruitment interview to participate in additional postnatal assessments (Page 5, Paragraph 1). We also added a study flow chart to illustrate the derivation of the final analytic sample (Supplemental Figure 1).

2. At the beginning of the introduction, it is mentioned that early cognitive... development is a key determinant of future educational and income achievement (line 54-55). It is not discussed how best to measure this and it should be discussed to what extent developmental outcomes at 1 year are sufficient for this purpose.

Response: Thank you for this important point. We have clarified in the Discussion that developmental assessment at 1 year captures early neurodevelopmental function but may not fully reflect later-emerging cognitive, behavioral, and academic outcomes and longer-term follow-up would allow examination of a broader range of developmental domains (Page 17, Paragraph 2).

3. The introduction mentions that PAE is more risky in low income settings (line 69) than high income settings and various potential protective factors in high income settings are mentioned. The role of additional risk factors in low income settings should also be mentioned e.g. binge drinking is known to be a common pattern in South Africa.

Response: Thank you for this helpful suggestion. We have revised the Introduction to mention the contextual differences in exposure patterns (Page 3, Paragraph 2). We expanded the Discussion and refer readers to Supplemental Table 2, which describes differences in binge drinking patterns across sites (Page 15, paragraph 2).

4. In the methods section on Study Populations (line 96-109), little background is given regarding the nature of the two study sites. Some further context would help readers to interpret the article e.g. are the residential areas urban vs rural, how do SES generally compare to average in each country?

Response: Thank you for this helpful sugges

---

## [Editor Report · Decision Letter 1]

6 Mar 2026

Maternal obesity and prenatal alcohol exposure are associated with child development: Results from the Safe Passage Study

PONE-D-25-11614R1

Dear Dr. Sania,

We’re pleased to inform you that your manuscript has been judged scientifically suitable for publication and will be formally accepted for publication once it meets all outstanding technical requirements.

Kind regards,

Emma K. Kalk

Academic Editor

PLOS One
---

## [Editor Report · Acceptance letter]

PONE-D-25-11614R1

PLOS One

Dear Dr. Sania,

I'm pleased to inform you that your manuscript has been deemed suitable for publication in PLOS One. Congratulations! Your manuscript is now being handed over to our production team.

Kind regards,

on behalf of

Dr. Emma K. Kalk

Academic Editor

PLOS One